# MODELING THE SECOND PLAYER IN DISTRIBUTIONALLY ROBUST OPTIMIZATION

**Paul Michel**
School of Computer Science
Carnegie Mellon University
pmichel1@cs.cmu.edu

**Tatsunori Hashimoto**
Computer Science Department
Stanford University
thashim@stanford.edu

**Graham Neubig**
School of Computer Science
Carnegie Mellon University
gneubig@cs.cmu.edu

## ABSTRACT

Distributionally robust optimization (DRO) provides a framework for training machine learning models that are able to perform well on a collection of related data distributions (the "uncertainty set"). This is done by solving a min-max game: the model is trained to minimize its maximum expected loss among all distributions in the uncertainty set. While careful design of the uncertainty set is critical to the success of the DRO procedure, previous work has been limited to relatively simple alternatives that keep the min-max optimization problem exactly tractable, such as $f$-divergence balls. In this paper, we argue instead for the use of neural generative models to characterize the worst-case distribution, allowing for more flexible and problem-specific selection of the uncertainty set. However, while simple conceptually, this approach poses a number of implementation and optimization challenges. To circumvent these issues, we propose a relaxation of the KL-constrained inner maximization objective that makes the DRO problem more amenable to gradient-based optimization of large scale generative models, and develop model selection heuristics to guide hyper-parameter search. On both toy settings and realistic NLP tasks, we find that the proposed approach yields models that are more robust than comparable baselines[1].

## 1 INTRODUCTION

Machine learning models trained with empirical risk minimization (ERM) are able to achieve high aggregate performance on data sampled from their training distribution. However, they often exhibit drops in accuracy when confronted with data from domains that are under-represented in their training data, such as those of different topic (Gururangan et al., 2020), sociolect (Blodgett et al., 2016), accent (Amodei et al., 2016) or writer age (Hovy & Søgaard, 2015) in language processing tasks, or skin color (Grother et al., 2019) or lighting (Georghiades et al., 2001) in image processing tasks. This is a particularly egregious issue in applications where higher error rates can have far reaching negative implications, such as the silencing of underrepresented minorities in toxicity detection systems (Dixon et al., 2018) or disparity amplifying feedback loops in credit rating models (Fuster et al., 2018).

This behaviour often arises from the objective function of ERM, where the parameters $\theta$ of the model are learned by minimizing the expectation of a loss function $\ell$ under a data distribution $p$ (or, specifically in practice, an associated empirical data distribution $\hat{p}$)

$$\mathcal{L}_{\text{ERM}}(\theta) = \mathbb{E}_{(x,y)\sim\hat{p}}\ell(x, y, \theta). \tag{1}$$

When the model encounters data sampled from a different distribution $q_{\text{test}} \neq p$, performance can suffer significantly. Distributionally robust optimization (DRO) (Ben-Tal et al., 2013b) provides a natural solution to this issue by replacing the expected risk under a single distribution $p$ with the *worst* expected risk over a pre-determined family of distributions $\mathcal{Q}$ (the "uncertainty set")

$$\mathcal{L}_{\text{DRO}}(\theta) = \max_{q\in\mathcal{Q}}\mathbb{E}_{(x,y)\sim q}\,\ell(x, y, \theta). \tag{2}$$

---

[1]Code to reproduce our experiments can be found at https://github.com/pmichel31415/P-DRO

If $\mathcal{Q}$ contains $q_{\text{test}}$, the DRO objective upper bounds the expected risk under $q_{\text{test}}$. However, *a priori* knowledge of possible test distributions is not always available or easy to acquire. For example, training a model to be robust to some demographic attributes ($\mathcal{Q} = \{q_{\text{demographic 1}}, q_{\text{demographic 2}}, \cdots\}$) requires collecting and annotating data with the necessary information, an expensive and ethically fraught endeavour. In the absence of such information, one has to resort to defining the uncertainty set analytically, drawing on one's intuition of what constitutes a possible test distribution given the observed training distribution, such as using moment constraints (Delage & Ye, 2010; Nguyen et al., 2020), $f$-divergence (Ben-Tal et al., 2013a; Hu & Hong, 2013; Faury et al., 2020), Wasserstein/IPM (Sinha et al., 2018; Husain, 2020) balls, or coarse-grained mixture models (Oren et al., 2019; Hu et al., 2018). However, the need for keeping the inner supremum in Eq. (2) tractable limits the possible choices.

In this paper, we propose that the uncertainty set be instead defined as a family of parametric generative models. The resulting DRO objective (§2) is a differentiable game with two players: the original model $\ell(x, y; \theta)$ and a model of its worst-case distribution $q_\psi(x, y)$, the titular "second player" which we hereafter refer to as *the adversary*. Using this formulation — which we call Parametric DRO (P-DRO) — allows for more flexibility in the choice of the adversary's architecture (and so the uncertainty set). Unfortunately, finding a solution of this game via direct application of simultaneous gradient descent (Singh et al., 2000) is difficult (Balduzzi et al., 2018). In particular, direct gradient descent on the uncertainty set suffers from instability due to the large variance of the gradients (Greensmith et al., 2004), and hyper-parameter selection is not straightforward.

To address these challenges, we make two main contributions (§3): first, we propose a new relaxation of the DRO game's inner maximization problem (with KL constraints). The resulting objective is more amenable to simultaneous gradient update than the original zero-sum game and significantly improves training stability, while still yielding useful adversaries. Second, we develop a principled approach for selecting hyper-parameters: we leverage the learned adversaries to decide which of any two given models trained with P-DRO is more robust than the other.

We do an in-depth set of experiments analyzing the effect of our proposed changes on both a toy task as well as a more realistic, yet still synthetic sentiment classification task (§4). Finally, we show that in the more realistic setting of toxicity detection, P-DRO yields models that are more robust to changes in demographic groups, even though these groups are unknown at training time, opening up applications in combatting dataset bias (§5).

## 2 PARAMETERIZING THE UNCERTAINTY SET

Consider a model parameterized by $\theta \in \mathbb{R}^{d_{\text{model}}}$. Minimizing the DRO objective described in Eq. (2) over the uncertainty set $\mathcal{Q}$ turns the optimization problem into the min-max (or zero-sum) game

$$\min_{\theta \in \mathbb{R}^d} \max_{q \in \mathcal{Q}} \mathbb{E}_{(x,y) \sim q} \ell(x, y, \theta). \tag{3}$$

The first player controls the parameters $\theta$, whilst the second player controls the worst-case distribution $q$. In the absence of explicit information on groups of interest (such as demographics, domain, etc.), an adequate choice of the uncertainty set $\mathcal{Q}$ is critical to the success of DRO. This is in fact very much an active area of research (Sinha et al. (2018); Duchi & Namkoong (2018); Oren et al. (2019), see Rahimian & Mehrotra (2019) for a survey). $\mathcal{Q}$ must be sufficiently large to contain test distributions of interest, but if it is too large it may contain "adversarial" distributions on which no model can perform well. Moreover, the design of $\mathcal{Q}$ is also circumscribed by the necessity of keeping the min-max problem tractable, particularly in the context of stochastic optimization. In Hu & Hong (2013) and Duchi et al. (2016) for example, the choice of $f$-divergence balls allows the use of duality arguments to reformulate (3) as a more manageable min-min problem. Others, like Hu et al. (2018) or Oren et al. (2019), propose using mixture models, the simplicity of which enables them to solve the inner maximization problem efficiently.

Instead, we propose to explicitly model the second player in the DRO game as a parametric model $q_\psi$ of the data. Of course, not all parameterizations $\psi \in \mathbb{R}^{d_{\text{adv}}}$ of a given generative model represent useful distributions, and we require that the adversary stay "close" to the underlying true data distribution $p$. As a measure of distance between $q_\psi$ and $p$, we choose the KL (Kullback & Leibler, 1951) divergence due to its wide acceptance in the machine learning community, as well as its appealing

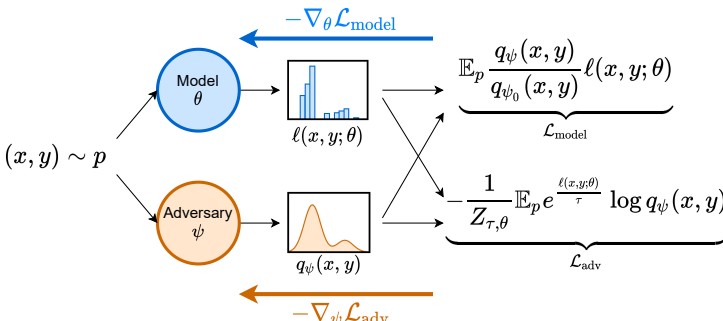

Figure 1: Summary of P-DRO: At every step of training, $(x, y)$ pairs are sampled from the data distribution $p$ and fed to both the model $\theta$ and the adversary $\psi$. For every sample, the model produces loss values $\ell(x, y; \theta)$ and the adversary produces densities $q_\psi(x, y)$. Both are combined into $\mathcal{L}_{\text{model}}$ and $\mathcal{L}_{\text{adv}}$, which are used to update the $\theta$ and $\psi$ respectively, via simultaneous gradient updates.

properties in the context of DRO.[2] The KL upper bound, $\kappa$, is left as a parameter to be decided by the experimenter. We refer to the resulting DRO formulation as Parametric DRO

$$\min_{\theta} \max_{\substack{\psi \\ \mathrm{KL}(q_\psi \| p) \leq \kappa}} \underbrace{\mathbb{E}_{(x,y) \sim q_\psi} \ell(x, y, \theta)}_{\mathcal{L}_{\text{P-DRO}}(\theta, \psi)} . \tag{4}$$

## 3 OPTIMIZING P-DRO

The min-max problem in Eq. (4) belongs to a class of games called "differentiable games" (another famous representative being generative adversarial networks (Goodfellow et al., 2014)). We can search for a solution of this game with simultaneous gradient descent (Singh et al., 2000), *i.e.* by simultaneously updating $\theta$ and $\psi$ with $-\nabla_\theta \mathcal{L}_{\text{P-DRO}}$ and $\nabla_\psi \mathcal{L}_{\text{P-DRO}}$ respectively. Unfortunately, in general, there is no theoretical guarantee that simultaneous gradient descent will converge to a Nash equilibrium[3] (Balduzzi et al., 2018), nor that any such equilibrium even exists if the objective is non-convex in $\theta$ (or non-concave in $\psi$). The success of GANs and the follow-up literature (Wang et al., 2019) serves as an encouraging example that gradient based methods can yield useful solutions despite the pessimistic theoretical results. In this section, we discuss difficulties that arise when optimizing $\theta$ and $\psi$ jointly, and propose modifications of the objective to address them.

### 3.1 TRAINING THE MODEL $\theta$

We could train the model $\theta$ by taking negative gradient steps on $\mathbb{E}_{(x,y) \sim q_\psi} \ell(x, y; \theta)$. This gradient can be estimated by sampling examples from $q_\psi$ and averaging the gradient of their losses. Unfortunately, this objective requires that $q_\psi$ is well-behaved at all iterations, as it is the only source of supervision for $\theta$. If $q_\psi$ is initialized incorrectly or begins producing unrealistic $(x, y)$, the quality of $\theta$ degrades as it begins to learn a predictor on invalid training examples from $q_\psi$. As an alternative, we opt to compute the gradients for $\theta$ with importance sampling, *i.e.* rewriting $\mathcal{L}_{\text{P-DRO}}$ as $\mathbb{E}_{(x,y) \sim p} \frac{q_\psi(x,y)}{p(x,y)} \ell(x, y; \theta)$, which ensures that all $(x, y)$ samples will be derived from the training set itself. Unfortunately, the true density $p$ is unknown to us. As an approximation, we replace $\frac{q_\psi(x,y)}{p(x,y)}$ with the likelihood ratio between $q_\psi$ and the maximum likelihood estimate of $p$, $q_{\psi_0} := \arg\max_{q_\psi} \mathbb{E}_{(x,y) \sim p} \log q_\psi(x, y)$. This changes the min-max problem to

$$\min_{\theta} \max_{\substack{\psi \\ \mathrm{KL}(q_\psi \| p) \leq \kappa}} \underbrace{\mathbb{E}_{(x,y) \sim p} \frac{q_\psi(x,y)}{q_{\psi_0}(x,y)} \ell(x, y, \theta)}_{\mathcal{L}_{\text{model}}} . \tag{5}$$

---

[2]For instance: $\mathrm{KL}(q \| p) < +\infty$ implies that $q$ stays within the support of $p$

[3]Nash equilibria (Osborne & Rubinstein, 1994) can be thought of the game theoretic analog of global minima in optimization.

This becomes a simple expected loss objective, which we can estimate by sampling from the empirical distribution $\hat{p}$. In experiments, we find that with this formulation we are able to train robust $\theta$ even when $q_\psi$ is only a mediocre generative model (see Appendix C.2). To further stabilize training at the beginning of the optimization process, we initialize $\psi$ with $\psi_0$, making the objective exactly the same as ERM for the first gradient step.

## 3.2 TRAINING THE ADVERSARY $\psi$

According to Eq. (5) the adversary $\psi$ must maximize $E_{(x,y) \sim q_\psi} \frac{p(x,y)}{q_{\psi_0}(x,y)} \ell(x, y, \theta)$ within a KL ball of fixed radius. This is challenging for several reasons: first, enforcing the bound is intractable for complex families of adversaries where *e.g.* projecting onto the KL ball is another difficult optimization problem of its own. Second, maximizing the expectation with respect to the parameters of the distribution $q_\psi$ is prone to instability due to large gradient variance (Greensmith et al., 2004).

**Lagrangian Relaxation**   To address the first difficulty, we loosen the strict KL constraint and instead consider the Lagrangian relaxation $\mathbb{L}$

$$\mathbb{L}(\psi, \tau) = \mathbb{E}_{(x,y) \sim q_\psi} \frac{p(x,y)}{q_{\psi_0}(x,y)} \ell(x, y, \theta) - \tau \left( \mathrm{KL}(q_\psi \| p) - \kappa \right). \tag{6}$$

We fix the Lagrangian multiplier $\tau > 0$ as treat it as a "temperature" hyper-parameter. With some reorganization (which we develop in Appendix A.1), we can show that

$$\mathbb{L}(\psi, \tau) = -\tau \mathrm{KL}(q_\psi \| q^*_{\tau,\theta}) + C. \tag{7}$$

Where $q^*_{\tau,\theta} \propto p(x,y) e^{\frac{p(x,y)}{q_{\psi_0}(x,y)} \frac{\ell(x,y;\theta)}{\tau}}$ and $C$ is a constant in $\psi$. In other words, maximizing $\mathbb{L}$ in $\psi$ is equivalent to minimizing the KL divergence between $q_\psi$ and $q^*_{\tau,\theta}$. One difficulty with this objective is that $q^*_{\tau,\theta}$ depends upon the unknown probability density $p(x, y)$. We avoid this problem by treating the density ratio $\frac{p(x,y)}{q_{\psi_0}(x,y)}$ as a constant, which is closely related to assumptions that have been used successfully in past formulations of DRO (Oren et al., 2019). Empirically, we find that incorporating $q_{\psi_0}$ as a surrogate for $p$ is a serviceable approximation, as demonstrated in Section 4.

**Reversing the KL**   Minimizing the KL divergence in this direction is difficult for several reasons. First, it entails optimizing an expectation in $q_\psi$ over $\psi$, which is difficult due to the large variance of the gradients (Greensmith et al., 2004). Second, computing this KL necessitates access to the true theoretical density $p(x, y)$ in order to compute $q^*_{\tau,\theta}(x, y)$ in the argument of the expectation, but this quantity is unknown in practice.[4] To sidestep these issues, we elect to minimize the reverse direction $\mathrm{KL}(q^*_{\tau,\theta} \| q_\psi)$ instead. Due to the KL divergence being non-symmetric, this is a rather crude approximation[5], the implications of which are discussed in Norouzi et al. (2016). However, we find that this approach dramatically stabilizes the gradient dynamics while still yielding good adversaries, as observed empirically in Section 4.4. Discarding the entropy term (constant in $\psi$), the resulting problem is equivalent to minimizing

$$\mathcal{L}_{\mathrm{adv}}(\psi, \tau) := -\frac{1}{Z_{\tau,\theta}} \mathbb{E}_p\, e^{\frac{\ell(x,y;\theta)}{\tau}} \log q_\psi(x, y) \tag{8}$$

in $\psi$, where $Z_{\tau,\theta} = \mathbb{E}_p\, e^{\frac{\ell(x,y;\theta)}{\tau}}$ is the normalizer of $q^*$. In this case, we can estimate this expectation by substituting the empirical distribution $\hat{p}$ for $p$ in the expectation.

**Computing the Normalizer**   Approximating the inverse normalizer $\frac{1}{Z_{\tau,\theta}}$ in a minibatch yields a biased estimator. On the other hand, computing $Z_{\tau,\theta}$ over the entire training data at each step is prohibitive since it requires computing the loss of every single example. As a middle ground, we keep a running normalizer $\tilde{Z}_k$ computed from the average of the normalizers over a fixed number

---

[4]Note that substituting the empirical distribution $\hat{p}$ for $p$ poses issues here, because $q_\psi$ is not absolutely continuous with respect to $\hat{p}$.

[5]For instance, the optimum of the reverse KL doesn't necessarily match that of the forward KL within the parametric confusion set $\mathcal{Q}$

$k$ of consecutive minibatches. In other words, if $B_i$ and $\theta_i$ denote the minibatch and adversary parameters at step $i$ respectively, the normalizer at step $t$ will be

$$\tilde{Z}_k = \frac{1}{\sum_{i=t-k}^t |B_i|} \sum_{i=t-k}^t \sum_{x,y \in B_i} e^{\frac{\ell(x,y;\theta_i)}{\tau}}. \qquad (9)$$

If $k$ is too low, there is a risk of under-estimating the normalizer, especially if the distribution of weights contains infrequent high weight samples. On the other hand, if $k$ is too high there is a risk of using "stale" weights in the normalizer. In experiments, we treat $k$ as a hyper-parameter.

### 3.3 OPTIMAL STOPPING

When should one stop training a model with P-DRO? In ERM it is customary to stop training after the empirical risk — periodically evaluated on a held out validation dataset — stops decreasing. This is particularly important to prevent over-fitting to the training data. However, it is not an appropriate criterion for P-DRO, since the model is not trained to minimize empirical risk in the first place. A more pertinent choice is to compare the robust validation losses

$$\mathcal{L}_{\text{robust,valid}}(\theta) = \max_{q_\psi \in \mathcal{Q}} \underbrace{\frac{1}{|D_{\text{valid}}|} \sum_{x,y \in D_{\text{valid}}} \frac{q_\psi(x,y)}{q_{\psi_0}(x,y)} \ell(x,y;\theta)}_{:=\mathcal{L}_{\text{valid}}(\theta,\psi)}. \qquad (10)$$

However, finding the inner supremum for each of the $T$ evaluation checkpoints $\theta_1 \ldots \theta_T$ is expensive as it requires solving $T$ independent optimization problems. Instead, we leverage the existence of adversaries $\psi_t$ associated with each model $\theta_t$, as well as the initial adversary $\psi_0$ and take the maximum over the $T+1$ adversaries $\{\psi_0, \ldots, \psi_T\}$. Since our relaxation of the P-DRO objective loosens the KL constraint, we need weed out adversaries which might violate it. Specifically, we estimate the $\text{KL}(q_\psi \| p) = \mathbb{E}_p \, q_\psi/p \log q_\psi/p$ on the validation set, using $q_\psi/q_{\psi_0}$ as a stand-in for $q_\psi/p$, and reject all adversaries for which the result is greater than a threshold, which we set to $\log 10$ based on preliminary experiments detailed in Appendix C.1.[6] We refer to this stopping criterion as **Minmax**.

Computing the full min-max necessitates keeping track of $T$ models and $T+1$ adversaries, which is ponderous when the model is large. As a solution, we propose an approximation, **Greedy-Minmax**, in which we only keep one best model $\theta^*$. At each evaluation step $T$, we compare $\theta_T$ to $\theta^*$, and update $\theta^*$ to whichever achieves lower robust validation loss over the $T+1$ adversaries $\psi_0, \ldots, \psi_T$.

By keeping track of only one additional model, and using the weights $\frac{q_{\psi_t}(x_i,y_i)}{q_{\psi_0}(x_i,y_i)}$ of individual examples in $D_{\text{valid}}$ as sufficient statistics for computing the loss against each adversary, Greedy-Minmax can be achieved with space complexity $2d_{\text{model}} + T|D_{\text{valid}}|$, which is much more efficient than the $T(d_{\text{model}} + d_{\text{adv}})$ of Minmax.

### 3.4 HYPER-PARAMETER SELECTION

Our proposed P-DRO method relies on 3 different hyper-parameters (in addition to the model's hyper-parameters): the adversary learning rate $\lambda$, the temperature $\tau$ and the size of the re-normalizing window $k$. As a consequence, we need a reliable criterion for deciding which of two configurations is better. This model comparison bears many similarities with the stopping problem described above. Therefore, we resort to a similar solution: given two models $\theta_1$, $\theta_2$ trained with P-DRO, and their respective adversaries $\{\psi_0^1, \ldots, \psi_T^1\}$, $\{\psi_0^2, \ldots, \psi_T^2\}$ (for instance, the adversaries associated with $\theta_1$ and $\theta_2$ at periodic checkpoints during training), we select the best model following

$$\theta^* = \underset{\theta \in \{\theta_1, \theta_2\}}{\arg\min} \; \max_{\psi \in \{\psi_0^1, \ldots, \psi_T^1, \psi_0^2, \ldots, \psi_T^2\}} \mathcal{L}_{\text{valid}}(\theta, \psi). \qquad (11)$$

---

[6]To simplify notation, this additional constraint is implicit in the rest of this section.

# 4 EXPERIMENTAL ANALYSIS OF P-DRO

Before moving on to a real world scenario in Section 5, we first demonstrate that P-DRO is able to learn robust models in a synthetic Natural Language Processing (NLP) task, and perform ablation studies to examine the importance of the various modifications described in Section 3.

## 4.1 EXPERIMENTAL SETTING

For analysis purposes, we design a simple NLP task amenable to DRO. We specifically choose NLP as a domain due to the striking success of language models as generative models of textual data (Sundermeyer et al., 2012; Radford et al., 2018), which can be used to model the uncertainty set. We base our task off of the binary version of the Stanford Sentiment Treebank dataset (SST-2; Socher et al. (2013)), which we modify to introduce spurious correlation. Specifically, we introduce a distractor token to some sentences. The distractor we use consists of prepending "so , " to the sentence ("i hated this movie" $\longrightarrow$ "so , I hated this movie"), which doesn't change the underlying sentiment. The resulting samples can be categorized in 4 "groups" depending on their label (positive or negative) and the presence or absence of the distractor. In particular, we add this distractor to 95% of the negative reviews and 5% of the positive reviews in the training and validation set, so that the presence of the distractor strongly correlates with negative sentiment (a similar construction is proposed in (Utama et al., 2020)). In the test data, we modify 50% of all sentences for each class equitably to ensure that there is enough data in each group, but we report "average" test accuracy by re-weighting the group accuracies to mimick the training distribution. We call this modified task **BiasedSST**.

For the classifier, we train a simple one layer BiLSTM model with embedding/hidden dimension 300. For the adversary, we adopt an auto-regressive transformer model based on the successful GPT-2 language model architecture but with 6 layers, a dimension of 512 and 8 attention heads (we experiment with a smaller, LSTM based adversary in Appendix C.2). In order to model the input output pair $(x, y)$, we pre-pend a special label-specific token to sentences before running them through the language model. We train the model with Adam (Kingma & Ba, 2014) and the adversary with vanilla stochastic gradient descent (which we found more stable in experiments). We refer to Appendix B for specific details of the experimental setting.

## 4.2 P-DRO CAN LEARN ROBUST MODELS

We train 7 models with P-DRO on BiasedSST using different hyper-parameters for the adversary. We start from configuration $\lambda = 10^{-4}$, $\tau = 0.01$, $k = 5$, and for each hyper-parameter we run a configuration with a smaller and a higher value, keeping all other hyper-parameters the same. We train for 50 epochs and select the best model using the strategies described in Section 3.

We also compare three other approaches. First, to appreciate how well the model could perform if the groups were known at training time, we train with Group-DRO on the oracle groups using an exponentiated-gradients based online algorithm (**Oracle DRO**; Sagawa et al. (2020)). Second, we implement **Topic CVaR** (Oren et al., 2019), a

Table 1: Average and robust accuracies on BiasedSST. Underlining indicates statistically significant difference compared to ERM ($p < 0.05$)

|  | Robust | Average |
|---|---|---|
| ERM | $2.15 \pm 0.97$ | $95.09 \pm 0.16$ |
| Topic CVaR | $\underline{5.18} \pm 1.46$ | $95.00 \pm 0.10$ |
| NonParam | $\underline{28.11} \pm 2.16$ | $\underline{92.45} \pm 1.55$ |
| P-DRO | $\underline{34.98} \pm 9.39$ | $\underline{84.21} \pm 2.11$ |
| Oracle DRO | $\underline{67.71} \pm 3.03$ | $\underline{77.91} \pm 4.49$ |

method for DRO on NLP where the uncertainty set is determined by mixtures of a topic model. Finally, we compare to non-parametric DRO with a Kullback-Leibler (KL) constrained uncertainty set (Hu & Hong, 2013; Hu et al., 2018), which we adapt to fit our online mini-batch training setting (**NonParam**). We refer to Appendix B.3 for details and hyper-parameters of the baselines.

We report the worst-case ("robust") accuracy over all groups on the test set, as well the average accuracy in Table 1 (we report the mean and standard deviation over 5 runs). We find that both Topic-CVaR, NonParam and P-DRO are more robust than ERM, but the latter outperforms the former two close to 30 and 7 points respectively, achieving $52\%$ of Oracle DRO's robust accuracy, while not leveraging any information on the oracle groups.

Table 2: Effect of different optimal stopping and hyper-parameter selection strategies on robust validation accuracy.

(a) Optimal stopping

| Criterion | Robust accuracy |
|---|---|
| Average | $0.00 \pm 0.00$ |
| Minmax | $25.22 \pm 13.01$ |
| +KL constraint | $31.30 \pm 10.07$ |
| +Greedy-Minmax | $\mathbf{32.17} \pm \pm 11.20$ |
| Oracle | $50.95 \pm 5.01$ |

(b) Hyper-parameter selection

| Criterion | Robust accuracy |
|---|---|
| Average | $31.03 \pm 12.16$ |
| Minmax | $28.62 \pm 12.37$ |
| +KL constraint | $\mathbf{35.65} \pm 11.47$ |
| Oracle | $38.26 \pm 13.01$ |

## 4.3 OPTIMAL STOPPING AND HYPER-PARAMETER SELECTION ABLATION

To understand the importance of the optimal stopping and hyper-parameter selection strategy described in Section 3.3, we perform an ablation on the BiasedSST dataset comparing 4 strategies:

- **Average**: models are selected based on their average zero-one loss (*i.e.* error rate) on the unmodified validation set. This is the baseline stopping criterion.

- **Minmax**: selection based on the adversaries (as described in Section 3.3), with and without the **KL constraint**, as well as its variant **Greedy-Minmax** for stopping.

- **Oracle**: in this setting the groups are known (in the validation set), and models are selected based on their error rate on the worst performing group. This is the optimal criterion for the group-DRO setting we are considering.

To compare stopping criterions experiments, we only consider one set of hyper-parameters: $\lambda = 10^{-4}$, $k = 5$ and $\tau = 0.01$. From the robust validation accuracies reported in Table 2a, we first observe that Average stopping results in a robust accuracy of 0, highlighting the necessity for a suitable stopping criterion. We find that Minmax, especially with a KL constraint, is a much better strategy, recovering $\approx 60\%$ of the performance achievable with Oracle stopping. Notably, the Greedy-Minmax variant which we use in practice reaches very close results ($< 1$ point difference) despite its requiring to keep track of only 2 out of the 50 model checkpoints at any time.

To understand the effectiveness of the Minmax strategy for selecting hyper-parameters. We take the models trained in Section 4.1, but select the best hyper-parameters using the different strategies described above. Results, shown in Table 2b, confirm that Minmax (with the KL constraint) is a better choice than Average for selecting hyper-parameters, even though the improvement is not as striking as for stopping.

## 4.4 IMPORTANCE OF $\mathcal{L}_{\text{ADV}}$

Finally, we investigate the importance of modifying the adversary's objective as described in Section 3.2. For this experiment, we devise a simpler toy task on which directly training the constrained DRO objective is possible. Specifically, we consider the two-dimensional binary classification problem pictured in Figure 2. The training data consists of 10,000 points partitioned in two normally distributed "domains" with a 1:50 sampling ratio and different classification boundaries. We train a logistic regression model, which cannot perfectly fit the training data and must trade-off between accuracy on each domain. For the sake of simplicity, we only model the input variables $x^7$ as isotropic normal distributions with fixed variance: the adversaries' parameter $\psi \in \mathbb{R}^2$ represents the location of the Gaussian (we fix the variance to the empirical variance of the data).

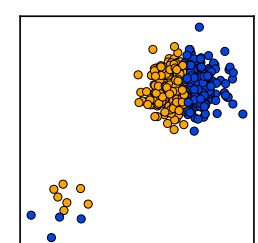

Figure 2: A toy classification task.

---

[7] In other words, we set $q_\psi(x, y) = p(y \mid x) q_\psi(x)$, where $p(y \mid x)$, is the true conditional which will be canceled out in the ratio $\frac{q_\psi(x,y)}{q_{\psi_0}(x,y)}$.

We compare 3 different versions of P-DRO: first, naive simultaneous gradient descent on the zero-sum game, without any constraint on the adversary (**bare P-DRO**), then the same, but with an approximation of the explicit KL constraint between $q_\psi$ and $q_{\psi_0}$ (**+KL constraint**; see Appendix A.2 for more details). Finally we report results using our relaxation and the KL reversal described in Section 3.2 (**+$\mathcal{L}_{adv}$**). For each setting, we report the average and robust accuracy with mean and standard deviation over 10 runs. For the KL constraint and the relaxation, we report the best results among 4 values of the KL bound $\kappa$ and the temperature $\tau$ respectively.

Table 3: Ablation of P-DRO to train the linear model on the toy task. We report accuracy on both domains, as well as robust accuracy.

|  | Average | Robust |
|---|---|---|
| ERM | **84.66** $\pm$ 0.10 | 49.75 $\pm$ 0.05 |
| bare P-DRO | 49.97 $\pm$ 0.10 | 49.85 $\pm$ 0.10 |
| +kl constraint | 76.63 $\pm$ 9.93 | 58.41 $\pm$ 9.25 |
| +$\mathcal{L}_{adv}$ | 76.97 $\pm$ 0.43 | **64.32** $\pm$ 1.31 |

In Table 3, we observe that bare P-DRO is too unstable and systematically diverges. The addition of a KL constraint mitigates this behaviour, but the zero-sum objective is still unstable, as evidenced by the high standard deviations. Finally, we find that the addition of $\mathcal{L}_{rev}$ stabilizes the training process greatly, leading to consistently high robust accuracy.

## 5 P-DRO IN PRACTICE: CASE STUDY OF TOXICITY DETECTION

In this section, we demonstrate the effectiveness of P-DRO in the more realistic setting of toxicity detection, the task of recognizing various forms of toxic language (eg. hate speech or offensive language). Identifying online abuse on the internet is a crucial challenge, and has garnered much interest in the NLP community (Schmidt & Wiegand, 2017; Fortuna & Nunes, 2018). However, recent work (Sap et al., 2019) has shown that there is strong correlation between toxic labels and the presence of certain markers of dialects of English spoken by minority groups. This correlation is in turn amplified by hate speech classifiers trained on such data, leading to biased prediction.

Our results on BiasedSST suggest that P-DRO can provide one solution to preventing models from absorbing spurious correlations present in their training data, even in the absence of protected attributes (such as language variety here).

### 5.1 EXPERIMENTAL SETTING

Following Sap et al. (2019) and Xia et al. (2020), we perform experiments on two datasets: **DWMW17** (Davidson et al., 2017), a corpus of 25K tweets classified in three categories: *hate speech* (6%), *offensive* (76%) and *neither* (18%), and **FDCL18** (Founta et al., 2018), a 100k sized dataset, also collected from Twitter and annotated with an additional *spam* label, with the following breakdown by categories: *hateful* (5%), *abusive* (27%), *normal* (54%) and *spam* (14%).

The released version of these datasets does not contain information on the dialect of each user. In order to be able to evaluate our models, and to train an Oracle DRO baseline, we follow Sap et al. (2019) and use annotations provided by the dialect classifier described in Blodgett et al. (2016) to label each example as one of four English varieties: White-aligned, African American, Hispanic, and Other. Note that, as these are automatically obtained labels, the groups may not exactly correspond to the actual racial sociolects, however Sap et al. (2019) does report that they correlate highly with self-reported race, and they serve as a useful proxy in the absence of manual annotation.

We formulate the group-DRO problem by separating each dataset into independent groups identified by both language variety and label, for a total of 12 and 16 groups for DWMW17 and FDCL18 respectively. Some of these groups are severely under-represented in the test set. In order to make our robust accuracy results reliable yet still representative of the under-represented groups, we combine groups that contain less than 100 samples into a single group to compute robust test accuracies.

On DWMW17, we train the same BiLSTM model as described in Section 4.3. To illustrate the applicability of P-DRO to other model architectures, we pick BERT (Devlin et al., 2018), a large scale pre-trained model as a classifier on FDCL18. In both cases, we adopt the Transformer architecture described in Section 4.3 as the adversary. We train the adversary with a temperature of $\tau = 0.01$ and a normalizing window $k = 10$. To demonstrate the efficacy of automatic hyper-parameter selection in the P-DRO setting, we delegate the choice of the adversary's learning rate $\lambda$ to grid-search, training 3 models with $\lambda \in \{10^{-5}, 10^{-4}, 10^{-3}\}$ and selecting the best using the Minmax criterion

Table 4: Robust test accuracy on the DWMW17 and FDCL18 toxicity detection tasks.

(a) No group information.

|  | DWMW17 | | FDCL18 | |
|---|---|---|---|---|
|  | Robust | Average | Robust | Average |
| ERM | $53.19 \pm 1.70$ | $69.44 \pm 0.53$ | $19.57 \pm 7.00$ | $\mathbf{81.56} \pm 0.26$ |
| Topic CVaR | $\underline{45.26} \pm 3.47$ | $\underline{61.68} \pm 5.02$ | $16.48 \pm 5.46$ | $\underline{80.49} \pm 0.49$ |
| NonParam | $54.13 \pm 1.14$ | $\mathbf{70.54} \pm 0.64$ | $\underline{17.54} \pm 6.41$ | $\underline{81.20} \pm 0.11$ |
| P-DRO | $\mathbf{69.06} \pm 1.70$ | $69.69 \pm 2.50$ | $\mathbf{30.25} \pm 10.13$ | $79.91 \pm 1.41$ |
| Oracle DRO | $\underline{74.50} \pm 1.74$ | $\underline{65.79} \pm 0.76$ | $\underline{55.23} \pm 3.97$ | $\underline{72.43} \pm 2.61$ |

(b) With group information on the validation set.

|  | Robust | Average | Robust | Average |
|---|---|---|---|---|
| ERM | $53.15 \pm 0.87$ | $69.64 \pm 1.01$ | $34.07 \pm 3.20$ | $78.78 \pm 0.38$ |
| Topic CVaR | $52.02 \pm 1.26$ | $\mathbf{69.11} \pm 0.49$ | $34.82 \pm 3.73$ | $\mathbf{79.59} \pm 0.85$ |
| NonParam | $49.41 \pm 5.60$ | $\underline{58.53} \pm 5.71$ | $43.13 \pm 6.97$ | $\underline{69.51} \pm 3.07$ |
| P-DRO | $\mathbf{63.05} \pm 4.25$ | $63.07 \pm 3.92$ | $\underline{47.61} \pm 4.53$ | $74.82 \pm 1.90$ |
| Oracle DRO | $\underline{74.50} \pm 1.74$ | $\underline{65.79} \pm 0.76$ | $\underline{55.23} \pm 3.97$ | $\underline{72.43} \pm 2.61$ |

described in Section 3.4. We also report numbers for Oracle DRO and Topic CVaR. Results are averaged over 5 runs, each with a different random seed.

## 5.2 CAN P-DRO PRODUCE MORE ROBUST MODELS?

Table 4a reports the robust test accuracies of all models on both tasks. Importantly, except for Oracle DRO, none of the methods compared here necessitate any knowledge of the groups, neither in the training nor validation data. We observe that in both settings P-DRO is able to achieve higher robust accuracy than ERM, Topic-CVaR and NonParam.

This suggests P-DRO as a useful option in case no group information whatsoever is available. However, in practice, it may be feasible to annotate at least a small amount of data with group information. To emulate this scenario, we perform the same experiment, but assume that group annotations are available on the validation data, which we use to determine optimal stopping and hyper-parameters. Results for this setting are reported in Table 4b. We find that, while the use of robust validation accuracy yields more robust models even for ERM (especially on FDCL18), P-DRO is still the best alternative that doesn't require group annotation on the training data.

## 6 IMPLICATIONS AND OUTLOOK

We have shown that there is promise in using parametric families of neural generative models for defining the uncertainty set in distributionally robust optimization. While we only perform experiments on NLP tasks, this approach can, in theory, be applied in any modality and in future work we hope to pursue this direction. In such cases where good quality generative models are unavailable, or such model cannot produce densities efficiently, an interesting direction would be to model the likelihood ratio $q_\psi/p$ directly. This alternative formulation poses different implementation challenges, and we leave it as a promising avenue for future research.

## ACKNOWLEDGEMENTS

The authors would like to thank the anonymous reviewers for their insightful feedback which helped improve the paper to its current version. In addition, this paper greatly benefited from discussion and feedback from various colleagues at CMU, in particular Chunting Zhou, Haohan Wang, Zachary Lipton and Zico Kolter. This work was supported by a Facebook Sponsored Research Award and by the DARPA GAILA project (award HR00111990063). The views and conclusions contained in this document are those of the authors and should not be interpreted as representing the official policies, either expressed or implied, of the sponsors.

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

## A  DERIVATIONS

### A.1  REORGANIZING THE LAGRANGIAN $\mathbb{L}(\psi, \tau)$

Let us write the Lagrangian $\mathbb{L}$ explicitly:

$$\mathbb{L}(\psi, \tau) = \mathbb{E}_{(x,y) \sim q_\psi} \frac{p(x,y)}{q_{\psi_0}(x,y)} \ell(x,y,\theta) - \tau \left( \mathrm{KL}(q_\psi \| p) - \kappa \right) \tag{12}$$

$$= \mathbb{E}_{(x,y) \sim q_\psi} \frac{p(x,y)}{q_{\psi_0}(x,y)} \ell(x,y,\theta) - \tau \, \mathbb{E}_{(x,y) \sim q_\psi} \log \frac{q_\psi(x,y)}{p(x,y)} + \tau \kappa \tag{13}$$

$$= \tau \, \mathbb{E}_{(x,y) \sim q_\psi} \log \left( \frac{p(x,y) e^{\frac{p(x,y)}{q_{\psi_0}(x,y)} \frac{\ell(x,y,\theta)}{\tau}}}{q_\psi(x,y)} \right) + \tau \kappa \tag{14}$$

$$= \tau(\kappa - \mathrm{KL}(q_\psi \| q^*_{\tau,\theta})) + \log \left( \mathbb{E}_{(x,y) \sim p} \, e^{\frac{p(x,y)}{q_{\psi_0}(x,y)} \frac{\ell(x,y,\theta)}{\tau}} \right) \tag{15}$$

This last step requires that the log moment generating function of $\ell$ under $p$ exist for $\tau$. In most scenarios we consider, $\ell$ is typically the negative log likelihood of a neural network model, which is generally bounded. Therefore the moment generating function is defined everywhere.

Note that the KL term is the only one dependent on $\psi$, therefore maximizing $\mathbb{L}$ for $\psi$ is equivalent to maximizing $-\mathrm{KL}(q_\psi \| q^*_{\tau,\theta})$, in other words minimizing $\mathrm{KL}(q_\psi \| q^*_{\tau,\theta})$

### A.2  ENFORCING THE KL CONSTRAINT IN THE TOY SETTING

Even in this simplest setting, the exact KL between $q_\psi$ (a gaussian) and $p$ (a mixture of gaussians) does not have an analytical expression (Hershey & Olsen, 2007). Instead, we fall back on enforcing the KL constraint between $q_\psi$ and $q_{\psi_0}$, both isotropic gaussians with the same standard deviation. Let $\mu$ and $\mu_0 \in \mathbb{R}^2$ denote their respective mean, and $\sigma > 0$ their standard deviation. In this context, their KL divergence reduces to:

$$\mathrm{KL}(q_\psi \| q_{\psi_0}) = \mathrm{KL}(q_{\psi_0} \| q_\psi) = \frac{1}{2\sigma^2} \| \mu - \mu_0 \|^2$$

In other words, the KL divergence is equivalent to the euclidean distance between the distributions' means. We use this fact to project $\psi$ (in the KL sense) onto $B_\kappa = \{ \hat\psi \mid \mathrm{KL}(q_{\hat\psi} \| q_{\psi_0}) < \kappa \}$:

$$\mathrm{proj}_{B_\kappa}(\psi) := \arg\min_{\hat\psi \in B_\kappa} \mathrm{KL}(q_\psi \| q_{\hat\psi})$$

$$= \psi_0 + \frac{\sqrt{2\kappa}\sigma}{\| \psi - \psi_0 \|} (\psi - \psi_0)$$

## B  EXPERIMENTAL DETAILS

We describe in more details some of the experimental settings for our NLP experiments. More details can be found in our code release: `https://github.com/pmichel31415/P-DRO`.

### B.1  MODEL SETTINGS

In all experiments, we split the text into sub-word tokens using the tokenizer described in (Devlin et al., 2018). During training, we sample minibatches that contain at most $64$ sentences or $2500$ tokens, whichever is greater, in order to prevent GPU memory overflow in case of long sentences.

We train all models with Adam (Kingma & Ba, 2014) with an initial learning rate of $2 \times 10^{-5}$, which we decay linearly at each step until the end of training. We validate the models every epoch. For BERT, we start from the `bert-base-uncased` checkpoint.

## B.2 ADVERSARY SETTINGS

In all experiments, we use a Transformer model based on the GPT-2 architecture (Radford et al., 2019) to serve as the adversary. In order to initialize the adversary (to obtain $\psi_0$), we first pre-train the model on a generic, relatively large language modeling dataset, WikiText-103 (Merity et al., 2017). We also use a batch size of 64 samples or 2500 tokens, and train with Adam for 10 epochs, with a fixed learning rate of $3 \times 10^{-4}$. Then, we fine-tune this model on each dataset, this time minimizing the negative log-likelihood of the $(x, y)$ pair (by introducing the special "[label]" token as described in Section B), using the same hyper-parameters but a smaller learning rate ($10^{-5}$). We find that, due to the small to medium size of the datasets under consideration, this LM pretraining step helped achieve lower error on the generative modeling task.

## B.3 BASELINE SETTINGS

### B.3.1 TOPIC CVAR

To train the topic model for Topic CVaR, we first pre-process the text by removing all punctuation, urls and user mentions (for twitter data). Importantly, we remove stop-words for our toxicity experiments but *not* for our BiasedSST experiment. This is because the distractor token we use ("so") belongs to most English stop words lists, and removing it would completely prevent the topic model from picking up on the groups of interest. We then estimate the parameters of the model with Gensim[8] and use similar settings as Oren et al. (2019) ($\alpha = 0.1$, $\beta = 1.0$), setting the number of topics to 10.

For both Oracle-DRO and Topic-CVaR, we use the algorithm proposed in Sagawa et al. (2020) to estimate the worst-case group (either oracle group or topic in Topic-CVaR) online during training. We perform grid-search over $\{1, 0.1, 0.01\}$ to find the best learning rate for the group weights update. For Oracle DRO, the best model is simply selected by robust validation accuracy. For Topic CVaR, unless specified otherwise, we select the model with the lowest worst-case error over all topics.

### B.3.2 NONPARAM

In the KL-constrained non-parametric setting, the min-max problem reads

$$\min_{\theta} \max_{\substack{q \text{ s.t.} \\ \text{KL}(q_\psi \| p) \leq \kappa}} \mathbb{E}_{(x,y) \sim q} \ell(x, y, \theta). \tag{16}$$

Here, $\kappa$ is the desired radius of the KL ball, and is treated as a hyper-parameter. The solution of the inner maximum has an analytical solution of the form $q_\theta^* = \frac{a}{Z_{\theta,\tau^*}} p(x, y) e^{\frac{\ell(x,y;\theta)}{\tau^*}}$ (see Hu & Hong (2013); Hu et al. (2018) for details) with $Z_{\theta,\tau^*} = \mathbb{E}_p e^{\frac{\ell(x,y;\theta)}{\tau^*}}$ and $\tau^*$ such that

$$\text{KL}(q^* \| p) = \mathbb{E}_p \frac{e^{\frac{\ell(x,y;\theta)}{\tau^*}}}{Z_{\theta,\tau^*}} \left( \frac{\ell(x, y; \theta)}{\tau^*} - \log Z_{\theta,\tau^*} \right) = \kappa.$$

Note that both computing $Z_{\theta,\tau^*}$ and $\text{KL}(q^* \| p)$ require taking expectations over $p$. In our setting, where $\ell(x, y; \theta)$ is the output of a large neural network, we cannot afford to take this expectation over the entire training data at each step. Instead, we fall back to taking the average over each mini-batch. We find $\tau^*$ with binary search in $\log_{10}$ space within the $[10^{-10}, 10^{10}]$ interval and clip to the lowest or highest value should the result lie outside the search interval.

---

[8] https://radimrehurek.com/gensim/

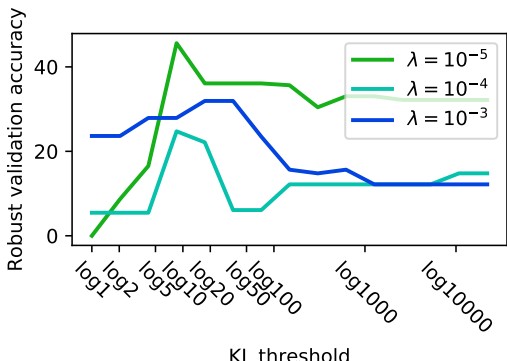

Figure 3: Evolution of the robust validation accuracy of the model selected by Greedy-Minmax as a function of the KL threshold $\kappa_{\text{valid}}$

In all experiments, we try 4 different values for $\kappa$: 0.01, 0.1, 1 and 10. Unless indicated otherwise, we perform early stopping and hyper-parameter selection using our Minmax criterion using the non-parametric weights as adversaries on the validation data.

## C ADDITIONAL EXPERIMENTS

### C.1 MINMAX VALIDATION KL THRESHOLD

The Monte-Carlo estimate of $\text{KL}(q_\psi \| p)$ on the validation set is $\frac{1}{|D_{\text{valid}}|} \sum_{x,y \in D_{\text{valid}}} \frac{q_\psi(x,y)}{p(x,y)} \log \frac{q_\psi(x,y)}{p(x,y)}$. Similarly to Section 3, we approximate the (unknown) likelihood ratio $\frac{q_\psi(x,y)}{p(x,y)}$ with $\frac{q_\psi(x,y)}{q_{\psi_0}(x,y)}$.

We want to reject all adversaries where this approximated KL is greater than some threshold, $\kappa_{\text{valid}}$, but how do we choose a good value for $\kappa_{\text{valid}}$? Consider an adversary which selects a fraction of the validation data of size $\alpha |D_{\text{valid}}|$ for some $\alpha \in (0,1]$. In such a case, the likelihood ratio is $1/\alpha$ on this subset and 0 everywhere else, and the resulting KL estimate will be $\log \alpha$. In other words, choosing a threshold of $\kappa_{\text{valid}}$ means allowing the adversary to potentially select any subset of size at least $1/e^{\kappa_{\text{valid}}}$ of the original data. Our heuristic choice, $\log 10$, corresponds to allowing subsets of size at least 10% of $|D_{\text{valid}}|$.

Of course, this is only a heuristic because the adversary can reweight the validation set non-uniformly. To assess the effect of $\kappa_{\text{valid}}$ on Greedy-Minmax, we compute the average robust validation error of the selected model across 5 runs for 3 different values of the adversary's learning rate. Results on BiasedSST, depicted in Figure 3, show that adversaries with higher learning rate are more sensitive to the choice of threshold, but all values of $\kappa_{\text{valid}}$ between $\log 5$ and $\log 20$ seem to work for these settings.

### C.2 P-DRO EXPERIMENTS WITH AN LSTM ADVERSARY

We replicate the experiments BiasedSST experiments in Section 4, but this time using a smaller generative model, which is unlikely to generate good samples. Specifically, we use a one layer LSTM model (Hochreiter & Schmidhuber, 1997) with embedding and hidden dimension 256. We only perform grid-search over $\lambda \in [10^{-5}, 10^{-4}, 10^{-3}]$ and select the best with Minmax.

Once pre-trained on the BiasedSST dataset, this model achieves a perplexity of 227.0, more than 4 times worse than the transformer model we use in other experiments (49.8). However, as evidenced by its robust accuracy displayed in Table 5, P-DRO is still able to learn a robust model. We take this as evidence that the re-weighting introduced in Section 3 helps stabilize training even when $q_\psi$ is not a perfect model of the data.

Table 5: Average and robust accuracies on BiasedSST when P-DRO is trained with an LSTM adversary. Underlining indicates statistically significant difference compared to ERM ($p < 0.05$)

|  | Robust | Average |
|---|---|---|
| ERM | $2.15 \pm 0.97$ | $95.09 \pm 0.16$ |
| Topic CVaR | $\underline{5.18} \pm 1.46$ | $95.00 \pm 0.10$ |
| NonParam | $\underline{28.11} \pm 2.16$ | $\underline{92.45} \pm 1.55$ |
| P-DRO | $\underline{43.68} \pm 4.93$ | $\underline{86.58} \pm 1.77$ |
| Oracle DRO | $\underline{67.71} \pm 3.03$ | $\underline{77.91} \pm 4.49$ |

Table 6: Effect of hyper-parameters on robust validation accuracy on BiasedSST

|  | Robust accuracy | |
|---|---|---|
|  | Minmax stopping | Oracle stopping |
| $\lambda = 10^{-5}$ | $28.62 \pm 12.37$ | $45.10 \pm 4.50$ |
| $\lambda = 10^{-4}$ | $44.74 \pm 3.24$ | $50.43 \pm 5.05$ |
| $\lambda = 10^{-3}$ | $25.57 \pm 10.33$ | $38.70 \pm 2.97$ |
| $\tau = 0.1$ | $39.72 \pm 5.55$ | $50.00 \pm 4.98$ |
| $\tau = 0.01$ | $44.74 \pm 3.24$ | $50.43 \pm 5.05$ |
| $\tau = 0.001$ | $44.74 \pm 3.24$ | $50.87 \pm 5.09$ |
| $k = 1$ | $41.98 \pm 4.48$ | $49.60 \pm 5.39$ |
| $k = 5$ | $44.74 \pm 3.24$ | $50.43 \pm 5.05$ |
| $k = 10$ | $32.17 \pm 11.20$ | $50.95 \pm 5.01$ |

## C.3 INFLUENCE OF HYPER-PARAMETERS ON P-DRO

We study the influence of the 3 hyper-parameters $\tau$ (temperature), $k$ (size of the renormalization window) and $\lambda$ (learning rate of the adversary) on the performance of P-DRO. All experiments are run on the BiasedSST dataset, and the analysis proceeds as follows: starting from configuration $\tau = 0.01$, $k = 5$ and $\lambda = 10^{-4}$ and vary each of the hyper-parameters independently. We report two numbers for each configuration: robust accuracy of the best model using Greedy-Minmax stopping and using Oracle stopping. The latter is useful to disentangle the effect of the stopping criterion.

As seen in the results shown in Table 6, we find that $\tau$ has the least effect on robust accuracies. While the renormalization window parameter $k$ has some effect on optimal stopping, the best robust accuracy achieved by the model (with oracle stopping) varies little. We observe the adversary's learning rate $\lambda$ to be the most sensitive hyper-parameter, which is why we restrict our grid-search to $\lambda$ in Section 5.

