# OpenReview forum: "Modeling the Second Player in Distributionally Robust Optimization"
_ICLR.cc/2021/Conference — ICLR 2021 Poster_

### Official Review · AnonReviewer4 · 2020-10-23
**Recommendation to Accept**

**Rating:** 7
**Confidence:** 4

**Review:**

This paper considers distributionally robust optimization (DRO) and uses the neural generative models to characterize the uncertainty sets. To tackle the optimization challenges, several implementation tricks are incorporated to solve the minimax problem. The proposed robust method is validated on NLP tasks.

This paper is well-written and of a good structure. Although the main idea is simple, the authors make several modifications to the algorithm to make it tractable and with performance guaranteed heuristically. To summarize, the main contribution of this paper is a new algorithm that combines standard techniques, such as Lagrangian relaxation and KL reverse, into the DRO problem with KL uncertainty sets. And this algorithm was shown to perform well under synthetic and real-data NLP tasks. Since there is no novel techniques proposed in this paper and there is no performance guarantee for the proposed framework, overall, I think this is a borderline paper due to its limitations in theoretical development and technical novelty.

Moreover, if the main focus of this paper is on developing a new computational framework that can lead to more robust results, then the authors should compare with more benchmark methods, while I only see the comparison with ERM, Topic-CVaR, etc. For example, I am wondering is it applicable to compare with Wasserstein DRO or Huber's classical work of Total variation based DRO, or some other DRO works in the literature, so that it will be more convincing on the performance of the proposed method.

A minor typo in the paper: in section 6, there is a duplicated "produce" in the sentence: "In such cases where good quality generative models are unavailable, or such model cannot produce produce densities efficiently".

---

> ### Author Response · Authors · 2020-11-13
> **Response to AnonReviewer4**
>
> We thank the reviewer for their encouraging comments and helpful feedback. We address their specific concerns below, and we are happy to continue discussing any of these points or answer follow-up questions.
>
>
> \> Since there is no novel techniques proposed in this paper and there is no performance guarantee for the proposed framework, overall, I think this is a borderline paper due to its limitations in theoretical development and technical novelty
>
> We do agree with the reviewer that the paper does not make a strong theoretical contribution, and our approach is very much motivated by proposing a method that works in a practical scenario rather than what is most satisfying theoretically.
>
> The main novelty of the paper is the use of a parametric family as the uncertainty set in DRO. However, our contribution goes beyond the brute-force approach of simply plugging parametric models into the classical DRO min-max (which doesn’t work, as demonstrated in our toy experiments in Section 4.4). In particular while a number of the adjustments detailed in Section 3 are not novel by themselves (lagrangian relaxation, KL reversal...), the fact that they can be combined and applied to the problem of parametric DRO is (in the authors’ opinion) far from being a given.
>
>
> \> More baselines
>
> The reviewer’s point regarding more baselines is well taken. First, we would like to point out that Wasserstein DRO presumes the existence of a canonical metric on the input space, of which there is none for discrete sequential inputs such as natural language sentences. Adaptation of Wasserstein DRO to NLP is an interesting direction, but it is far from straightforward, and would warrant a more thorough investigation of its own. Huber’s work on robust statistics solves a related but different setting: ensuring that models are robust to an adversary who modifies the training data. Our paper considers the problem where the training data is fixed, and the test distribution is potentially different. To our knowledge, Huber’s robust statistics approaches do not directly address KL-robust DRO problems.
>
> That being said, we did run the additional baseline of non-parametric KL-constrained DRO, inspired by the formulation of Hu et al. (2016) (https://arxiv.org/abs/1611.02041). We refer to our general response to all reviewers for more details and initial results. We are currently working towards adding these additional baseline results throughout the paper.

---

### Official Review · AnonReviewer1 · 2020-10-24
**Two key issues left unaddressed**

**Rating:** 6
**Confidence:** 4

**Review:**

TL;DR: The paper makes an interesting contribution from a practical point of view, but two important theoretical concerns need to be addressed in the rebuttal for acceptance.

The paper proposes the use of ideas taken from the literature on distributionally robust optimization within a parametric framework. More precisely, the main idea is to consider only a (parameterised) subset of the traditional KLD-uncertainty sets. As this avoids the need for elegant analytic solutions (at the expense of a more brute force computation), it has the flavour of a more ‘black box’ approach towards the deployment of DRO.  Overall, I really enjoyed the way this paper was written. Purpose and use of the contributions are clear throughout, and the reader is drawn in.  I also liked the contribution and believe that the paper demonstrated its ideas to be useful. There are however two points of major concern from a more theoretical side. In my mind, these are rather substantial, and I will list them below. To recommend that the paper be accepted, these points will have to be addressed in a future version of the paper:

(1) How do you ensure that the KLD between $q_{\psi}$ and $p$ is finite? p clearly is the empirical measure (as is emphasised e.g. just above eq. (5)), but $q_{\psi}$ will be continuous. This means that the KLD between the two distributions is not defined/infinity for any value of \psi (Mismatch of support problem). These kind of problems are the precise reasons why other quasi-distances (like the Wasserstein distances) have become increasingly interesting for ML. As far as I can tell, this problem is not elaborated upon anywhere in the paper.

(2) It is totally unclear to me why it should be viable to suddenly flip the direction of the KLD. The KLD is not symmetric and in general will not even have the same minimum. In fact, generally speaking the only time the minimum will be the same in either direction is when the KLD’s global minimum is such that $q_{\psi} = q_{\tau, \theta}$ (i.e. we can drop the KLD term for the loss in (7) completely, so that it simply equals $C$). Given the definition of $q_{\tau, \theta}$, it is unreasonable to assume that this global minimum is attained. This makes the flipping of the KLD’s direction questionable at best. Calling the outcome an ‘approximation’ is then grossly inaccurate.  (See e.g. the visualisations here: https://wiseodd.github.io/techblog/2016/12/21/forward-reverse-kl/)

Lastly, since the chief concern of the paper is the construction of new uncertainty sets, I would have liked to see two additional recent references discussed which have produced uncertainty sets purely based on moments (https://arxiv.org/abs/2007.04458, ICML 2020) as well as on general IPMs (https://arxiv.org/abs/2006.04349, NeurIPs 2020). Both these types of uncertainty sets do *not* suffer from the mismatch of support problem, and—like the famous f-divergence based uncertainty sets—have elegant dual forms.


POST-DISCUSSION: The authors promised to clarify the two issues I pointed out in ways that are satisfactory for a paper whose main concern is practicality (as opposed to theoretical rigour). I will thus raise my score to a weak accept.

---

> ### Author Response · Authors · 2020-11-13
> **Response to AnonReviewer1**
>
> We appreciate the reviewer’s enthusiasm for our approach, and are grateful for the insightful feedback. We address their specific concerns below, and we are happy to continue discussing any of these points or answer follow-up questions.
>
> \> How do we ensure that the KL divergence between q_\psi and the empirical distribution p is finite or even well-defined?
>
> The reviewer makes an keen observation that the empirical distribution, being of finite support, does not have a finite KL divergence with q_\psi. In truth, we are only interested in the KL divergence between q_psi and the true underlying data distribution, which we can reasonably assume is finite. We agree that the phrasing of the paper is confusing in this regard, as we interchangeably refer to both the “true” data distribution and the empirical distribution (of finite support over the training data) as p. We will edit the paper to make this clearer.
>
>
>
> \> Why is flipping the KL viable?
>
>
> The reviewer is correct that the KL is not symmetric, and as such the reversed loss L_rev is not equivalent to the original “forward” KL minimization problem. First, we would like to clarify that we did try to optimize the forward-KL constrained objective and found in our toy experiments (Section 4.4) that this generally failed. This failure is echoed by a variety of previous work (eg. RAML (Norouzi et al., 2016), but also in the RL literature). We do agree that flipping the KL divergence is an unsatisfactory approximation (and we will update the paper to further emphasize this point), however as shown empirically in previous work, it seems to be effective in practice.
>
> Ultimately, we choose to make concessions to the optimization concerns, to the expense of theoretical exactness. We do believe that attempting to directly minimize the forward KL is a promising future direction. However, the current version of the paper demonstrates empirically that the KL reversal is not only viable, but is also sufficient for P-DRO to yield more robust models. If performing P-DRO with the forward KL results in superior results than the results that we have obtained with reverse KL, then we argue that it would only further improve the utility of our already-promising approach.
>
> \> Missing references (Husain, 2020 and  Nguyen et al. 2020)
>
> We thank the reviewer for pointing out these recent relevant references, which we’ll include in the upcoming revised version of the paper.

---

> > ### Comment · AnonReviewer1 · 2020-11-17
> > **Thanks + follow-up question**
> >
> > I thank the authors for the extensive answer and appreciate the time they took!
> >
> > (1) Regarding the finiteness of the KL, I am afraid your comment does not address my concern completely: Even though your KL-ball is well-defined between true data-generating distribution and the model (provided that both admit densities that are absolutely continuous with respect to one another), there is a remaining problem. Specifically, since you have no access to the *true* data generating mechanism, you will have to approximate the KL-ball with the empirical data distribution for computation. If I understand correctly, eq. (7) will still depend on the empirical distribution ($p(x,y)$) via $q_{\tau, \theta}^*$. This means that the problem would persist---unless you only evaluate $q_{\psi}$ at the finitely many support points of $p(x,y)$. What do you do in practice? Whatever you do, it should go into the paper :)
> >
> > (2) I have to say that the arguments for reversing the KL are really unsatisfactory. Of course it's not your fault that the KL behaves badly when optimized, but it raises the question why you would like to define your objective the way you do. Could you not reverse the direction of the KL in your uncertainty set so that it directly appears in the 'correct direction' in eqs (6) and (7)? I don't see any part of your argument chain that would prevent you from doing that, and saying that "optimizing the forward direction is hard" would directly justify why you are defining the uncertainty set based directly on the reverse KL? Please let me know if this would be impossible, but I don't see why it would be.

---

> > > ### Author Response · Authors · 2020-11-18
> > > **Answer to follow up questions**
> > >
> > > \> I thank the authors for the extensive answer and appreciate the time they took!
> > >
> > > Likewise, the reviewer’s willingness to engage in discussion and clarify their concerns is much appreciated.
> > >
> > > \> (1) Regarding the finiteness of the KL, [...] What do you do in practice? Whatever you do, it should go into the paper :)
> > >
> > > In practice, we reverse the $KL(q_\psi || q_{\tau, \theta}^*)$ to $KL( q_{\tau, \theta}^* || q_\psi)$ which can be re-written  ($\mathcal L_{rev}$, Eq. 8) as the ratio of $\mathbb E_p e^{\ell/\tau}\log q_\psi$ and the normalizer $Z_{\tau,\theta}=\mathbb E_p e^{\ell/\tau}$. Both are expectations (over $p$) of random variables which do not depend on $p$ (and hence can be computed directly). In both cases, we indeed plug-in the empirical distribution in place of $p$ in the expectation. This will be clarified in the revision (in Section 3.2 after Eq. 8) by adding the following sentence: “In practice, we compute the expectation by sampling from the empirical distribution.”.
> > >
> > > \> (2) I have to say that the arguments for reversing the KL are really unsatisfactory. [...] Could you not reverse the direction of the KL in your uncertainty set so that it directly appears in the 'correct direction' in eqs (6) and (7)?
> > >
> > > This is an interesting remark. In fact, defining the uncertainty set based on the reverse KL would lead to similar optimization issues, at least without some additional approximations or tricks. The reason for this is that the resulting lagrangian ($\mathbb E_{q_\psi} \ell + \tau E_p \log q_\psi + [\text{Constant in }\psi]$) still contains a term in $\mathbb E_{q_\psi}$ which is difficult to optimize for $\psi$. On the other hand, the reversal of the KL in Eq 7 (leading to the $\mathcal {L}_{rev}$ objective in Eq. 8) allows us to avoid taking an expectation on $q_\psi$ which is the main optimization hurdle. Because of this, our formulation is not directly equivalent to reversing the direction of the KL constraint on the uncertainty set.

---

> > > > ### Comment · AnonReviewer1 · 2020-11-20
> > > > **response**
> > > >
> > > > Thanks for the swift response!
> > > >
> > > > Regarding the second point, that makes sense. I am still dissatisfied with the reasons for suddenly flipping the KL because it fundamentally changes the optimization problem, but I don't have to die on this hill.
> > > >
> > > > For the first point however, I am not sure you are understanding my concern. If you use the empirical measure, then this means that $p_{\tau, \theta}^*$ is supported only on finitely many points. This means that the KL that you *actually* comptue in practice is
> > > > $$ \sum_{i=1}^n q_{\psi}(x_i) \log(q_{\psi}(x_i) / p_{\tau, \theta}^*(x_i)),$$
> > > > i.e. you discretize the density $p_{\tau, \theta}^*$ into a new measure supported only on (finitely-many) support points $x_i$. Defining this discrete measure (for the dirac delta $\delta_x(y)$ being $0$ everywhere except if $y=x$) as
> > > > $$ m_{\psi}(x)=1/n\sum_{i=1}^n\delta_{x_i}(x) \cdot q_{\psi}(x) $$
> > > > this means the uncertainty set that you are *actually* computing is with respect to
> > > > $$\text{KL}(m_{\psi}\|p_{\tau, \theta}^*) \neq \text{KL}(q_{\psi}\|p_{\tau, \theta}^*)$$
> > > > Note that this is not some stickler remark: If the two measures $\nu$ and $\mu$ are not absolutely continuous with respect to one another, then (by definition!) we have that $\text{KL}(\mu \|\nu) = \infty$. In other words, while $\text{KL}(m_{\psi}\|p_{\tau, \theta}^*)<\infty$, the mismatch of support problem means that $\text{KL}(q_{\psi}\|p_{\tau,\theta}^*) = \infty$.
> > > >
> > > > For a definition of the KL in the measure-theoretic sense, see e.g. Def. 359 here:
> > > > https://www.google.com/url?sa=t&rct=j&q=&esrc=s&source=web&cd=&ved=2ahUKEwjW2aHA-pDtAhU7QkEAHbTYBWwQFjANegQIYxAC&url=https%3A%2F%2Fwww.stat.cmu.edu%2F~cshalizi%2F754%2F2006%2Fnotes%2Flecture-28.pdf&usg=AOvVaw1tPO-Fq79f_PqetUPJuYfD

---

> > > > > ### Author Response · Authors · 2020-11-23
> > > > > **Clarification**
> > > > >
> > > > > Again, we thank the reviewer for taking the time to clarify their concern.
> > > > >
> > > > > We will attempt to rephrase the order of operations in Section 3.2, and what we believe to be the reviewer’s core issue. First, for clarity’s sake, we define the following notation
> > > > >
> > > > > - $p$: the true data distribution and $m$ the empirical distribution
> > > > > -  $q^*=q_{\tau,\theta}^*=\frac 1 Z p e^{\ell/\tau}$ and $m^*$ its restriction to the empirical distribution
> > > > > - $q_\psi$: the adversary and $m_\psi$ its empirical counterpart (as defined by the reviewer)
> > > > >
> > > > > In our intended presentation, Section 3.2 proceeds as follows:
> > > > >
> > > > > 1. Write the lagrangian relaxation $-E_{q_\psi}\ell + \tau KL(q_\psi || p) + Constant = KL(q_\psi || q^*) + Constant$
> > > > > 2. Reverse the order of the KL: $KL(q^* || q_\psi)$
> > > > > 3. Plug-in the empirical distribution: $KL(m^* || m_\psi)$
> > > > >
> > > > > In particular in the last step, the plug-in of the empirical distribution on the left argument of the KL is standard practice and does not raise issues of absolute continuity. Insofar as one accepts the KL reversal, we believe that the use of the empirical distribution should be acceptable in this formulation.
> > > > >
> > > > > On the other hand, if we understand correctly, the reviewer reads our order of operation as:
> > > > >
> > > > > 1. Write the lagrangian relaxation $-E_{q_\psi}\ell + \tau KL(q_\psi || p) + Constant = KL(q_\psi || q^*) + Constant$
> > > > > 2. Plug-in the empirical distribution $KL(m_\psi || m^*)$
> > > > > 3. Reverse the order of the KL: $KL(m^* || m_\psi)$
> > > > >
> > > > > In this case we agree with the reviewer’s assessment that the transition from 1 to 2 is problematic because of absolute continuity issues. In fact, as far as we can see, there is no easy way to estimate the “correct” KL $KL(q_\psi || q^*)$ without running into the aforementioned issues with the empirical distribution.
> > > > >
> > > > > Everything considered, the two derivations yield the same final objective (step 3., equation 8 in the paper), so the discussion ultimately comes back to the KL reversal. As we have argued above (and shown empirically in the paper), this approximation, while unsatisfactory, still allows us to train robust models in a tractable fashion.
> > > > >
> > > > > Finally, we agree that this discussion is important, and we will strive to make it clear in the paper, but we don’t think that it discounts the general idea of the paper, nor the experimental results.

---

> > > > > > ### Comment · AnonReviewer1 · 2020-11-23
> > > > > > **response**
> > > > > >
> > > > > > Thank you for the extensive response --- As stated in the original review, I agree that the general idea still makes sense and that the paper demonstrates its usefulness.
> > > > > >
> > > > > > I am delighted that the authors aim to include the explicit distinction between empirical and theoretical KL-uncertainty set because it is of both theoretical and practical relevance.
> > > > > >
> > > > > > Finally, with this I will also note that the authors have addressed the two concerns I had. Including a more critical discussion of the KL-reversal together with a more explicit notational distinction between idealized and empirical data distributions will fix my main issues with the paper.

---

### Official Review · AnonReviewer3 · 2020-10-28
**The paper proposes a novel and important DRO method, and good experiments are conducted to evaluate the efficacy of the proposed method.**

**Rating:** 7
**Confidence:** 3

**Review:**

The paper proposes to define the uncertainty set in the DRO problem as a family of parametric generative models, which is to allow more flexibility in the choice of the uncertainty set architecture. To realize this idea, the paper first proposes a new relaxation of the DRO game's inner maximization problem (with KL constraints) so as to improve the training stability. It then develops a principled approach to select the hyper-parameters of the proposed method.

Strengths:
+ The paper is well-written.
+ The proposed method is novel and important for the DRO community.
+ Experiments with real-world problems are conducted to evaluate the effectiveness of the proposed method. I particularly like the experimental analysis the authors conducted to understand the behavior of their proposed method.

Weaknesses:
- The experiments are only on NLP tasks.

I have few questions to the authors:
1) How good the adversary model needs to be for the proposed method to perform well? In the experiments, an auto-regressive transformer model based on the GPT-2 language model is employed. What is the accuracy of this model on the train dataset of the DRO problem? Will the proposed method performance be too sensitive to the accuracy of the adversary model?
2) In the experiment (last paragraph of Section 5.1), the temperature \tau and the normalizing window k are fixed whilst the adversary learning rate \lambda is searched by grid-search. So how \tau and k are selected in practice? What is the performance of the proposed method when \tau and k vary?

---

> ### Author Response · Authors · 2020-11-13
> **Response to AnonReviewer3**
>
> We thank the reviewer for their encouraging feedback. We address their specific concerns below, and we are happy to continue discussing any of these points or answer follow-up questions.
>
> \> The experiments are only on NLP tasks
>
> While in general our proposed approach can be applied to any modality, the reviewer is correct that we only experiment on NLP datasets (except in our toy experiment in Section 4.4). As mentioned in the paper, this is motivated by the widely recognized success of language models, which make them a prime candidate for testing P-DRO.
> For other modalities where generative models are either not readily available or can’t provide normalized probabilities efficiently, such as GANs, an alternative solution might be to model the likelihood ratio q_psi/p directly, however this poses a variety of other challenges, which we defer to future work.
>
> \> How good of a generative model is the adversary, and how important is its performance?
>
> As pointed out by the reviewer, in most of our experiments, the adversary is a transformer model based on the GPT-2 architecture (albeit with fewer parameters than the actual GPT-2 model). On the BiasedSST dataset, this model attains a “perplexity” of 49.84 (note: this model predicts both label and text, as such the perplexity is not directly comparable to regular language models). Measuring the effect of the adversary’s performance on the effectiveness of P-DRO is an interesting ablation study. Should time and computing resources permit, we will make our best efforts to obtain additional results with smaller adversaries during the rest of the rebuttal period.
>
> \> How are \tau and k chosen in practice?
>
> As shown in Section 4, \tau and k can be chosen via grid-search using the Minmax criterion described in Section 3. For the experiments in Section 5 specifically, we fixed \tau and k in order to reduce the search space and make grid search more manageable. Possibly, better results could be obtained in Section 5 by searching for better \tau and k. We will edit the paper to clarify this.
>
> As to the effect of the choice of k and \tau on performance, we performed an ablation study on BiasedSST. We start from the configuration \lambda=10^-4, k=5 and \tau=0.01 and vary either k or \tau. We report two numbers for each configuration: robust accuracy of the best model using Greedy-Minmax stopping and using Oracle stopping. The latter is useful to disentangle the effect of the stopping criterion.
>
> ||Robust Accuracy (Minmax stopping)|Robust Accuracy (Oracle stopping)|
> |-|-|-|
> |k=0 | 41.98 ± 4.48 | 49.60  ±  5.39 |
> |k=5 | 44.74  ± 3.24 | 50.43 ± 5.05 |
> |k=10 |  32.17 ± 11.20 | 50.95 ± 5.01 |
> |-|-|-|
> |\tau=0.1 | 39.72 ± 5.55 | 50.00 ± 4.98 |
> |\tau=0.01 | 44.74  ± 3.24 | 50.43 ± 5.05 |
> |\tau=0.001 |  44.74  ± 3.24 | 50.87 ± 5.09 |
>
> Interestingly, neither k nor \tau have a strong effect on robust performance when using Oracle stopping. We will add these ablation studies to the updated manuscript.

---

> ### Author Response · Authors · 2020-11-20
> **Importance of adversary's accuracy on performance: results and discussion**
>
> We would like to add additional comments with regards to the reviewer's enquiry as to the importance of the adversary's accuracy:
>
> \> 1. How good the adversary model needs to be for the proposed method to perform well? In the experiments, an auto-regressive transformer model based on the GPT-2 language model is employed. What is the accuracy of this model on the train dataset of the DRO problem? Will the proposed method performance be too sensitive to the accuracy of the adversary model?
>
> We have replicated the experiments on BiasedSST, using a simple, one-layer LSTM model as the adversary. As it turns out, we had run these experiments already and they were in fact included in the original submission (albeit in the appendix; C.2). The setup is almost exactly the same as that of Section 3, except we only search for the learning rate in order to reduce the number of experiments (moreover, as outlined in our previous response, we find that the other hyper-parameters k and \tau have limited effect).
>
> This adversary achieves lower generative modeling performance than our autoregressive transformer (test perplexity 227.01 vs 49.84) on the biasedSST dataset. Yet, we find that it allows P-DRO to achieve robust test accuracy of 43.68 on biasedSST, which also outperforms all baselines.
>
> In fact, this is even higher than our results obtained with the transformer model. We find that this surprising result can be explained as an effect of the minmax validation strategy. When we remove this factor and choose hyper-parameters and early stopping with robust accuracy instead, we find that the LSTM adversary reaches a robust accuracy of 45.68 versus 47.53 for the transformer adversary, giving the latter a slight edge.
>
> In summary, we find that the size of the adversary as a generative has a limited effect when restricted to neural models. We will clarify the discussion in the paper and make sure to attract more attention to these results in the main text.

---

### Official Review · AnonReviewer2 · 2020-11-02
**Modeling the Second Player in Distributionally Robust Optimization**

**Rating:** 7
**Confidence:** 3

**Review:**

Good points
----
- The objective of the paper is sound: fight distributional shift in systems whose predictions
might have life-changing consequences (e.g data bias toxicity prediction models, etc.).
- The paper is well-written and easy to follow.

Bad points
----
- I don't see just how this model is "parametric". In statistics, "parametric" the adversarial
distribution is modeled as a gaussian, etc. with sought-for parameters (mean, covariance, etc.).
In the absence of that, I would have expected "parametric" to mean parametrizing the adversarial
distribution as the (softmax) output of a neural network. Neither of the above is the case in
this paper. So, what are the "parameters" in the proposed DRO adversary ? All I can see is that
the authors do  a full search over all distributions, subject to a KL constraint (see sections
2 and 3.2).
There is nothing "parametric" about this.
- The authors say "In particular, direct gradient descent on the uncertainty set suffers from
instability due to the large variance of the gradients (Greensmith et al., 2004), and
hyper-parameter selection is not straightforward." I'm not sure about this claim (which
is one of the main premises of the manuscript. What do the authors make of this paper
for example Faury et al. (AAAI 2020) "Distributionally Robust Counterfactual Risk Minimization" ?
The authors of that paper demonstrate how to efficiently formulate and solve KL-based DRO
problems. That paper also contains both theoretical and practical insights.
- The technical contribution of the paper is negligible (if any).
- The arguments in the paper very heuristic.
- Since the paper is supposed to be empirical (see previous points), I would have expected
experiments on real datasets.


Errors
---
- Change "solve the inner-max efficient" to "solve the inner maximization problem efficiently"
- Change "$x, y ~$ " to "$(x,y) ~ $" all through the manuscript
- Eqn (5): why not take $p$ and $q_{\psi_0}$ to equal the empirical distribution (as is usually
done) in DRO ?
- In eqn defining $q_{\psi_0}$, replace $\arg\max_{q_\psi}$ with $\arg\max_\psi$

---

> ### Author Response · Authors · 2020-11-13
> **Response to AnonReviewer2**
>
> We thank the reviewer for their detailed feedback. We address their specific concerns below, and we are happy to continue discussing any of these points or answer follow-up questions.
>
> \> I don't see just how this model is "parametric"
>
> The proposed approach is parametric in the sense that the confusion set is represented by a parametric family of models. To take the reviewer’s example, in one of our ablation experiments (Section 4.4), the adversary is indeed a gaussian, and its sought-for parameters are the mean.
> In our other experiments, the uncertainty set is composed of transformer-based language models. This setting is also parametric in the sense that each possible test distribution in the uncertainty set is associated with a set of parameters for a transformer model, and these parameters are optimized jointly with the classification model following the procedure described in Section 3.
>
> To clarify this point, we have added results for a non-parametric KL-constrained baseline. Please refer to our general response to all reviewers for more details.
>
> \> Comparison to Faury et al. (AAAI 2020) "Distributionally Robust Counterfactual Risk Minimization"
>
> We thank the reviewer for pointing out this relevant citation, which we’ll include into the next revision. As far as we can tell, this paper proposes a non-parametric KL-constrained formulation of DRO which is very similar to that of Hu & Hong (2013) or Hu et al. (2016), but applied to CRM. In fact, Faury et al. corroborate our point: they state (eg. Section 2.2) that  “We are interested in DRO instances that are amenable to direct optimization. To this end, we focus here only on uncertainty sets U based on information divergence”.
> In our work, we consider the challenges that occur when moving outside this tractable set of uncertainty sets, and consider intersections of the KL uncertainty set with parametric models where the inner-maximization problem becomes intractable (hence the need for the approximations described in Section 3).
>
> Again, we refer to our general response for more details on an additional, relevant baseline.
>
> \> The technical contribution of the paper is negligible (if any)
>
> To the best of our knowledge, we are the first to investigate DRO with neural-network based parametric confusions sets and to address the associated challenges (intractability of the inner-max, difficulty of enforcing the KL constraint...). We believe these are all technical contributions that are not attested to by previous research, and all required a significant amount of thought, design, implementation, and empirical validation.
>
> \>Since the paper is supposed to be empirical (see previous points), I would have expected experiments on real datasets.
>
> We would like to point out that the experiments in the final section of the paper are performed on two toxicity detection datasets, which are well established datasets addressing an important real problem and widely used in the community: Davidson et al. (2017) and Founta et al. (2018)  (760 and 109 citations respectively according to google scholar).
>
> \> In Eq. why not take q_psi_0 and p to equal the empirical distribution (as is usually done) in DRO ?
>
> Due to its parametric nature, the support of the adversary q_\psi is larger than that of the empirical distribution, therefore, we need to use the true data distribution p (which we assume has the same support as q_psi) in the denominator. In practice, since p is unavailable, we resort to the MLE q_\psi_0, which also has the same support.

---

> > ### Comment · AnonReviewer2 · 2020-11-25
> > **Final comment**
> >
> > Thanks to the reviewers making the effort of responding to the reviewers' concerns in-depth. The additional experiments were also a good idea. One reservation though: the gains of the proposed methods compared to the baselines (see response to all reviewers) seems too high. This kind of disproportionate performance is usually due to one of two things: (1) An issue a specific regime where the proposed methods are particularly better than the rest, and / or (2) A bug in the evaluation pipeline / protocol. I'd advise the authors to double-check. In the benefit of doubt, I'm increasing my score to 7.

---

### Author Response · Authors · 2020-11-13
**General Rebuttal**

We thank all four reviewers for their feedback. We address each reviewer’s specific concerns in separate replies, and are happy to continue discussing any of these points or answer follow-up questions.

A few reviewers brought up the comparison to the non-parametric KL-constrained approach which is similar to our proposed approach. We ran additional experiments to compare to this approach, inspired by the formulation of Hu et al. (2016) (https://arxiv.org/abs/1611.02041). We slightly adapted the algorithm to our setting (minibatch training of large models, we will outline those modifications in more detail in the upcoming revision of the paper). In particular, we experiment with 4 values for the radius of the KL ball (which controls the size of the uncertainty set): 0.01, 0.1, 1 and 10.

We report initial results on BiasedSST for two variants:

Average: we use average accuracy for both stopping and hyper-parameter selection
Minmax: we adapt our proposed Minmax criterion for stopping and hyper-parameter selection.

Results are as follows (robust test accuracy):

| Method| Robust Accuracy |
| --|--|
|ERM | 2.15  ± 0.97|
|Topic CVaR | 5.18  ± 1.46|
|Non-param (Average) | 8.51  ± 4.62|
|Non-param (Minmax) | 21.68  ± 4.85|
|P-DRO  | 34.98  ± 9.39|
|Oracle DRO | 67.71  ± 3.03|

First, we confirm that P-DRO yields more robust models than its non-parametric counterpart. Second, this further confirms the effectiveness of our proposed Minmax validation criterion, which also significantly improves the results of the non-parametric model.

We are currently working towards adding these additional baseline results throughout the paper.

---

### Author Response · Authors · 2020-11-24
**Final revision**

We again thank all four reviewers for their feedback, and in particular R1 for the insightful discussion.

We updated the paper to incorporate the reviewers’ comments. Here is a summary of the changes:

- Added another baseline: non-parametric KL-constrained DRO. This is in response to R3’s comments on the lack of baselines, and also intended to clear up confusions between parametric and non-parametric approaches to DRO brought up by R2
- Clarified the presentation to address R1’s concerns, specifically by:
    - Making the distinction between empirical and theoretical distribution more explicit where necessary, in particular in how it relates to estimating the KL divergence
    - Including a more nuanced discussion of the KL reversal
- Included additional experiments in the appendix to visualize the effect of various hyper-parameters, as suggested by R3.
- Highlighted appendix C.2 (experiments with a smaller adversary) better in the main text (in response to R3’s comment)
- Included additional references suggested by reviewers: Faury et al. (AAAI 2020; R2), Husain, 2020 and Nguyen et al. 2020 (R1)
- Fixed minor typos/presentation issues brought up by R2 and R4
- Adjusted a number (Greedy-minmax 30.43 -> 32.17) in Table 2a after a minor bug was fixed in our analysis code.

---

### Decision · Program_Chairs · 2021-01-07
**Final Decision**

**Decision:**

Accept (Poster)

**Comment:**

# Paper Summary

This paper considers the problem of distributionally robust optimization (DRO), in which one is attempting to minimize a loss on the worst of all distributions that are some distance (here, measured in terms of KL divergence) from the training set. The main novelty here is that this adversarial distribution is represented as a model, with parameters that are learned jointly with the primary model.

This is an intuitive idea, but as the authors explain, attempting to implement it leads to a number of complications. One of these is that it is challenging to constrain the adversarial distribution model to be a certain KL divergence away from the training set. To address this, they write down the Lagrangian, but do not actually optimize over the Lagrange multiplier resulting from this constraint: instead, they keep it at a fixed constant value (a hyperparameter). A second, and potentially more worrisome, issue is that it is difficult to optimize the KL divergence as written--instead, they swap the two parameters, which is of course incorrect but they claim leads to much nicer convergence behavior.

They also propose a stopping condition, which terminates optimization once the robust validation loss (i.e. the validation loss w.r.t. the worst permissible distribution) stops decreasing. Normally, this would require a search for the worst such distribution at every iteration, which would be prohibitively expensive, so they propose instead only checking the distributions that have been found by the adversary during the course of optimization.

They close with a set of experiments that is nicely designed to narrow in on and explore particular details of their approach (e.g. they have an experiment that validates their stopping criterion), and have a realistic experiment on two NLP datasets.

# Pros

1. Reviewers agreed that it was very well-written, well-organized, and comprehensive
1. Good discussion of background material. The paper is very accessible
1. Intuitive idea, although the details of the approach become somewhat complex
1. Aside from the "realistic" experiment, each is designed to explore a particular facet of their approach

# Cons

1. Some reviewers were concerned that the baselines were insufficient. In response the authors added the new Hu et al. baseline (NonParam), which seemed to be satisfactory
1. While the approach is more general, one reviewer noted that the experiments only consider NLP problems. This is a minor negative point, in my view
1. One reviewer was concerned that the results were "too good", and encouraged the authors to double-check their results. My belief is that, at least on the non-"realistic" experiments (which were mostly intended to drill down into specific attributes of their approach, rather than demonstrate its overall performance), this is because the problem was constructed to perform especially poorly with a non-DRO approach
1. One reviewer was unsatisfied with the idea of swapping the parameters to the KL divergence (I share this concern). The authors clarified, both in the response and in the paper, that swapping the parameters is indeed incorrect, and may in fact be a very bad approximation to the true quantity of interest, but that the performance difference was so dramatic that it couldn't be undone. This seemed to partially satisfy the reviewer

# Conclusion

All four reviewers ultimately recommended acceptance. The major concerns were (i) that the baselines weren't good enough (which the authors addressed by adding a new baseline), and (ii) that swapping the parameters to the KL divergence results in a very poor approximation to the original KL divergence (which the authors now explicitly acknowledge in the paper, with an explanation for why they feel it is necessary). Overall, this is a nice idea, and while bringing it into practice may require more hand-waving than would be ideal (which is the main reason I suggested a poster acceptance instead of a spotlight or oral), it seems to work well experimentally, and the experiments are overall very careful and well thought-out. Additionally, the writing quality is excellent, as is the organization and presentation of background material.